# Risk Assessment of Benzene, Toluene, Ethyl Benzene, and Xylene Concentrations from the Combustion of Coal in a Controlled Laboratory Environment

**DOI:** 10.3390/ijerph16010095

**Published:** 2018-12-31

**Authors:** Masilu Daniel Masekameni, Raeesa Moolla, Mary Gulumian, Derk Brouwer

**Affiliations:** 1Occupational Health Division, School of Public Health, University of the Witwatersrand, Parktown 2193, Johannesburg, South Africa; derk.brouwer@wits.ac.za; 2Department of Geography, Environmental Management and Energy Studies, University of Johannesburg, Aukland Park 2006, Johannesburg, South Africa; 3School of Geography, Archaeology and Environmental Studies, University of the Witwatersrand, Private Bag X3, WITS 2050, South Africa; raeesa.moolla@wits.ac.za; 4National Institute for Occupational Health, National Health Laboratory Services, Braamfontein 2001, Johannesburg, South Africa; mary.gulumian@nioh.nhls.ac.za; 5Haematology and Molecular Medicine, School of Pathology, University of the Witwatersrand, Parktown 2193, Johannesburg, South Africa

**Keywords:** coal, BTEX, hazardous air pollutants, domestic fuel burning

## Abstract

A D-grade type coal was burned under simulated domestic practices in a controlled laboratory set-up, in order to characterize the emissions of volatile organic compounds (VOCs); namely, benzene, toluene, ethylbenzene, and xylenes (BTEX). Near-field concentrations were collected in a shack-like structure constructed using corrugated iron, simulating a traditional house found in informal settlements in South Africa (SA). Measurements were carried out using the Synspec Spectras GC955 real-time monitor over a three-hour burn cycle. The 3-h average concentrations (in µg/m^3^) of benzene, toluene, ethylbenzene, p-xylene, and o-xylene were 919 ± 44, 2051 ± 91, 3838 ±19, 4245 ± 41 and 3576 ± 49, respectively. The cancer risk for adult males and females in a typical SA household exposure scenario was found to be 1.1 and 1.2 respectively, which are 110- and 120-fold higher than the U.S. Environmental Protection Agency (EPA) designated risk severity indicator (1 × 10^−6^). All four TEX (toluene, ethylbenzene, p-xylene and o-xylene) compounds recorded a Hazard Quotient (HQ) of less than 1, indicating a low risk of developing related non-carcinogenic health effects. The HQ for TEX ranged from 0.001 to 0.05, with toluene concentrations being the lowest, and ethylbenzene the highest. This study has demonstrated that domestic coal burning may be a significant source of BTEX emission exposure.

## 1. Introduction

The introduction of several chemicals into the atmosphere has been widely associated with increased health risks [1,2]. Anthropogenic sources of higher exposure to air pollutants are suggested to be attributed to industrial activities [3,4]. Several studies have been conducted globally, investigating the emissions of larger industrial activities such as power generation on the external environment [5,6]. The mechanisms as to how pollutants are emitted and distributed are well understood, especially on larger stationary sources in developed countries and parts of developing Asia. 

Globally, there is a growing concern regarding pollutant inventories in order to understand the major sources of emissions and their impacts [7]. There is an emerging body of knowledge which suggests that indoor household burning presents a major threat to public health [8,9] arising from lack of access to clean energy sources, which has been identified as a major contributor to local indoor air pollution [10,11]. The majority of households, especially in developing countries, rely on multiple energy sources combusted daily, using inefficient devices in poorly ventilated environments [12,13]. 

The emission of volatile organic compounds (VOCs) under these conditions may present an important class of pollutants as it has been associated with several health and environmental impacts [14,15,16]. It is reported that VOCs, even at low concentrations, can produce several health effects, including nausea, eye, and throat irritation, the induction of asthma attacks, fatigue, dizziness, and mental confusion [17,18,19,20,21]. VOCs in general are quite numerous; however, emphasis is given to mono-aromatic volatile organic compounds termed BTEX (benzene, toluene, ethylbenzene, and xylenes). This group of VOCs are often considered carcinogenic [22,23]. Particularly, benzene and ethylbenzene exposure is linked with an increased risk of leukemia and hematopoietic cancers [24,25,26]. Toluene and xylene are non-carcinogenic, but they may produce reproductive adverse effects; especially when exposures are chronic at low to high concentrations [27].

Efforts to create an exposure inventory for BTEX is mainly done in occupational environments, while less information is available in non-occupational settings [28,29,30,31,32,33,34]. The sources of BTEX in residential areas are diverse, including domestic care products; lifestyle-related chemicals such as cigarette smoke; and combustion energy-related sources [35]. It has also been suggested that the risk of exposure is higher in indoor environments relative to the outdoor environments [36,37,38,39,40,41]. 

Exposure to airborne pollutants is influenced by many factors, such as the emission rate at the source, air exchange rate, pollutant concentration and time spent indoors, and the meteorological conditions [32,35,38,42]. Children and the elderly are the most vulnerable groups, as they spend most of their time indoors and also due to a weaker immune system [42]. The study conducted by [32,35,43], have emphasized that infants and children are at greater risk than adults, due to their high metabolic and resting rate compared to adults. It was further found that children spend most of their time indoor next to their mothers, and they are thus exposed to elevated concentrations of combustion pollutants during cooking and heating conditions [20].

In regulating exposure to toxic compounds on human health, many countries use risk assessments as a tool to determine the relative risk, and to develop action plans based on emissions or concentration. However, a risk assessment considers various factors in estimating the possibility of a biological response. Factors such as hazard source identification, exposed group, exposure pathway, the concentration of the contaminant, target organ, and potential biological response dose, which might trigger a response, are investigated [32,42]. Hematotoxicity and immunotoxicity have been widely used as indicators for the non-carcinogenic effects of benzene exposure [44,45,46]. Chronic exposure to benzene have been reported in several studies, and reviews indicating the risk of anemia, bone marrow hyperplasia, aplastic anemia, leukopenia, lymphocytopenia, thrombocytopenia, and pancytopenia have been shown [24,46,47].

Exposure to high concentrations of BTEX have been widely associated with several adverse health effect in countries such as USA, India, and China [46,47,48,49,50]. Despite several human health effects reported elsewhere regarding exposure to BTEX, in South Africa, very few studies have been conducted to quantify indoor and environmental exposure to BTEX, especially from domestic activities where coal burning has been consistently linked to severe health effects [51,52]. The present study aims to quantify the concentrations of BTEX from domestic coal burning process, and evaluate the potential health risks with respect to cancer and non-cancer effects. The study uses experimental data on BTEX emission as proxies for near-field concentration, to estimate exposure mimicking the indoor use of coal in a brazier applicable in the South African informal settlements. Similar, studies conducted in this field mainly focused on the effect of fire ignition and ventilation on particulate matter and gaseous emissions (PM_2.5_ and PM_10_) [13,53]; while the study by [54,55] investigated the effect of coal particle size and moisture on gaseous and particulate matter, respectively. The presentation of data as an emission factor provides little information on the concentration of the studied compounds, and further, it become difficult to use such information in health-related studies.

The selection of the combustion device (high-ventilation stove) and the fire ignition method (top-lit updraft) is based on the South African government air pollution interim reduction strategy that is applicable for informal settlements. The combustion of coal in a highly ventilated stove, ignited by using the top-lit updraft ignition method, has been reported in several studies, to reduce particulate emissions by margins by up to 80%, compared to a low ventilated stove that is lit by using the bottom-up draft ignition method (BLUD) [13,53,56]; while no significant difference on the gaseous emissions were found in studies by [13,54]. Presently, epidemiological studies in South Africa use emission factor data to associate the exposure to health outcomes [51]. Consequently, the use of an emission factor as a concentration has been interchangeably used as the same term in several studies in these field, which might complicate the interpretation of results [54,51]. However, despite such reductions reported on PM, very few studies have been carried out to investigate the emissions of gases, especially VOCs, due to their inherent health risks. Therefore, in this study, BTEX emissions are used as proxies to determine the concentration that can be used to determine the dose in different exposure scenarios.

The study hopes to assist in contributing to knowledge on domestic solid fuel burning technologies, and it might aid in supporting future epidemiological and other studies in South Africa, and in other low-to-medium income countries with domestic coal burning activities, using similar combustion technology. In addition, it may be noted that this is the first study in a South African context that attempted to carry out a risk assessment on BTEX exposure that is applicable for informal settlement, according to the knowledge of the author. 

## 2. Materials and Methods

### 2.1. BTEX Sampling Condition

BTEX compounds were sampled under laboratory conditions simulating community-based activities. The combustion laboratory was constructed by using corrugated iron and combustion materials included coal, wood kindling, and paper. The selection of the stove, known as brazier (*imbaula*), to fuel combination (top-lit updraft (TLUD) and high ventilated stove), was based on the government project roll-out program of the TLUD ignition method as an interim air pollution reduction strategy initiative, and the selection of high ventilated stoves was based on local studies which proved that the use of a high ventilated stove lit with TLUD leads to the reduction of emissions [13,53]. Tests were performed over a period of three hours, and further details on the burn sequences are provided in [13,54]. The stove was lit by using the TLUD method in a high ventilated brazier. Further, details on the stove and fuel combination can be obtained in published literature, as contained in the references [13,16,56,57]. 

The study was carried out at the University of Johannesburg’s Sustainable Energy and Research Centre in South Africa. The stove and the GC955 sampling inlet were placed at the center of the combustion hut, respectively. The sampling location was made to mimic common practice with informal dwellers, where the stove is placed in the middle of the hut. The combustion laboratory was built to simulate a typical informal house, colloquially known as a shack, constructed using corrugated iron, with a small window (300 mm × 400 mm) and a standard door (840 mm × 1.8 m), as shown in Figure 1.

### 2.2. Domestic Combustion Scenario in a South African Low-Income Settlement

Before stove ignition, all openings leading to the outside of the shack were closed/sealed, mimicking field-based practices. Nevertheless, it must be noted that air leaks could occur, since the sealing of openings were not comprehensive enough to contain all emitted pollutants, which might be similar to a typical shack. The stove was placed at the center of the combustion lab, and measurements were taken 1 m above the floor, and 1.2 m away from the stove, as shown in Figure 1. A domestic coal fire is generally associated with high heat generation, simultaneously increasing the indoor temperature significantly. Due to the sensitivity of the monitoring equipment, care was given to separate the experimental and the data capturing rooms. The detection device was placed in the analysis room next to the combustion laboratory. The sampling probe, 1.9 m in length, was used to draw the exhaust to the detection device/gas analyzer. The isolation or removal of the detection device from the hot environment was done to avoid similar challenges experienced during field monitoring in [34], where the higher temperature led to the instrumentation malfunctioning, and a loss of data. 

Samples were taken and averaged for each distinct time aggregate (15 min, 45 min, and two hours, to coincide with burn cycles). The first sample was taken from the time the fuel was lit until the establishment of the flame, i.e., the first 15 min of the combustion where the condition is smoldering (i.e., burning slowly with visible smoke but without flames) with insufficient air supply and a low fuel bed temperature; the next stage is when the fire is well-established and the combustion process is at the mixing stage and takes about 45 min; the last stage where there is no visible flame, and only coke/fixed carbon is burning, and char formation often takes place (~120 min). The laboratory experiments were done three times per combustion time interval, where the average concentrations over three experiments were used in the study.

### 2.3. BTEX Sampling Instruments

In the present study, five VOCs were monitored using Synspec Spectras gas chromatography (GC955, series 600, Groningen, Netherland).This instrument is widely used to monitor BTEX, and has been approved as per the service specification EN 14662-3. The samples were drawn in through the inlet feeder, operated at a flowrate of 5 mL/min, connected at the back of the instrument. A 37 mm filter was connected between the monitoring instrument and the inlet probe, to isolate or exclude foreign particles. Drawn in hydrocarbons are firstly pre-concentrated in the Tenax GR, where they were pre-heated and desorbed, and thereafter separated according to the columns. The instrument is coupled with a photoionizer detector (PID) that assists in increasing the sensitivity for benzene and other aromatic hydrocarbons. The running cycle can be from 15 min upwards, which can be adjusted and operated at a temperature of <70 °C. Helium was used as a carrier gas, and set at a pressure of 350 kPa, connected using Teflon tubing at a distance of 1.8 m (from the gas cylinder to the GC).

### 2.4. Risk Assessment

BTEX emissions were monitored at near-field (inside the room, 1.2 m away from the emitting source), and they were then used as room concentrations. In a typical coal-burning house in South Africa, coal stoves are often used indoors during the winter season between 18:00 and 21:00. This is due to the inherent heat production of the technology, while in summer, it is unlikely that the stove will be used indoors for either cooking or space heating. The winter period in South Africa is from June to August (equivalent to 92 days). During winter, all of the outlets leading to the external environment are closed, with the intention to contain all produced or radiated heat from the device. Due to the hazardous nature of coal and the associated carbon monoxide poisoning, households often extinguish the fire and remove the stove from the indoor spaces before going to bed [57]. We therefore used this scenario to estimate the exposure duration.

In general, we have conducted a risk assessment to estimate the potential exposure to BTEX from domestic coal burning, and similar to other studies, we used the data to assess the risk to human health [48,58,59,60]. Risk assessment is a comprehensive process which includes hazard source identification, evaluation, characterization, and control, aiming at the prevention of possible health outcomes. Thus, this research has aimed at assessing both carcinogenic and non-carcinogenic potential risk by estimating the exposure to BTEX. 

#### 2.4.1. Hazard Identification

During data collection, BTEX concentrations from the coal burning device were monitored. The focus of the monitoring was limited to BTEX emissions, due to their inherent carcinogenic and non-carcinogenic health impacts on the exposed groups. As mentioned earlier, benzene is classified as a Group A human carcinogenic, according to U.S. Environmental Protection Agency (USEPA) [61,62]. Using the approach similar to the one described below, a lifetime exposure duration of 30 years was assumed for residential dwellers as a default value, based on the USEPA [62]. The common occupants in South African informal settlements are often unemployed individuals, or those who falls in the low income brackets. In the study conducted by [63,64], it was found that majority of the households live on an average income of R3500. Predominately, adult males and females reside in informal settlements, while there is only a small number of school-going children [65]. Informal settlements are often built closer to employment sites or in industrial zones [64,66]. The average age of dwellers was estimated to be 35 years, while the majority of shack owners are males, at 61% relative to females [65]. A small proportion of children below 18 years were recorded during a census in 2010, at 1% of the total dwellers in the informal sector [66].

#### 2.4.2. Exposure Assessment

For the exposure assessment, we have considered an estimated dose, expressed as a chronic daily intake (mg/kg/day). Due to the inadequate available methodologies for determining the internal dose, we used a near-field breathing zone concentration for the exposure assessment. We assumed that the breathing zone concentration is equal to the near-field concentration, or the emission zone [62]. The driving factors in dose estimation were the exposure pathway (air), including the route of entry (inhalation), the frequency to which one is expected to be exposed, the duration of the exposure, and the population age group (Adults male and females). Since this was a laboratory based study simulating the experience of residents, where information on population demographics is not present, the study adopted some of the parameters for the exposure scenario from the USEPA’s risk assessment guidelines, and South African Statistics as in Table 1 [62,65].

A dose-response relationship was used to estimate the potential biological response for each pollutant. Similar to [67,68], the average concentration for the entire burn cycle was used to calculate the chronic intake concentration. The chronic daily intake (CDI) (Equation (1)) for both the carcinogenic and non-carcinogenic pollutants was calculated by using the values as shown in Table 1. The average CDI_year_ provides the estimated daily intake corresponding to an annual dose.

For the exposure assessment, we have considered the estimated dose, expressed as a chronic daily intake (mg/kg/day). Due to inadequate available methodologies for determining the internal dose, we used a near-field breathing zone concentration for the exposure assessment. We assumed that the breathing zone concentration is equal to the near-field concentration or emission zone [62]. The driving factors in dose estimation were exposure pathway (air), including the route of entry (inhalation), the frequency to which one is expected to be exposed, the duration of exposure, and the population age group (adult males and females). Since, this was a laboratory-based study simulating the experience of residents, where information on population demographics is not present, the study adopted some of the parameters for the exposure scenario from the USEPA’s risk assessment guidelines and South African Statistics, as in Table 1 [62,65]:(1)$$CDI\left(\mathrm{averaged}\;\mathrm{daily}\;\mathrm{intake}\right)=\frac{C\times CF\times IR\times ED}{BW\times AT}$$

The chronic daily intake (CDI) determination was used as a basis for risk assessment calculation, similar to the current risk assessment studies [67,69,70,71], where:

CDI is the chronic (averaged) daily intake over a year (mg/kg/day);

C is the breathing zone concentration of BTEX in (µg/m^3^), derived from three identical experiments taken over a 3-h burn cycle;

CF is the concentration conversion (mg/µg = 0.001 or 1 µg) factor;

IR is the inhalation rate (default in adults, 20 m^3^/day);

ED is the exposure duration as in Equation (2) (11.5 days);

BW is the average body weight (70 kg, 60 kg for male and female adults, respectively);

AT is the number of days per year.

However, the default values as contained in Table 1 assume a daily intake of a pollutant over a 24 h period, is often constant, and can be extrapolated over a year. In our study, there was a variation on the exposure duration, due to the nature of how households use the technology.

In Equation (2), we determined a procedure that was used to estimate the exposure duration in a typical winter period in South Africa. The limitation of this method was that the exposure duration seeks to be confined to the coal combustion period (3 h), without taking into account the exposure resulting from accumulated concentrations that might take time to vent from indoor to outdoor. Since this was a laboratory study, the authors intentionally left out other variables in an ordinary house in informal settlements. Such variables may include the ventilation rate, or the building envelope, which influences the air ratio, taking into account the exchange from inside to outside. The exposure duration obtained in Equation (2) indicates a daily average exposure, given that the exposure involves a 3-h duration over a 92-day period in a year from this source (to allow for a full season).
(2)$$\mathrm{ED}=\frac{\mathrm{Actual}\;\mathrm{exposure}\;\mathrm{duration}}{24\;h\;}\times \;92\;\mathrm{days}$$
where:

ED is the exposure duration (days/year);

Actual exposure duration is the 3-h combustion period;

24 h is the total hours in a day; and 

92 days is the number of exposure days in a year.

In Equation (1), an average annual chronic daily intake was determined. However, for the risk assessment, a cumulative lifetime exposure concentration intake needs to be completed. In Equation (3), the average 30 years chronic dose (CDI_30 year_) is calculated by using the 30 year residential exposure duration, as obtained from USEPA default value:(3)$$CDI\left(30\;\mathrm{years}\;\mathrm{dose}\right)=\sum \frac{CDI\;\times \;365\;\times \;YE}{60/67}$$
where:

CDI is the cumulative average 30-year dose (mg/kg/day); 

CDI is the chronic daily intake (mg/kg);

YE is estimated lifetime residential exposure duration, which is equivalent to 30 years;

365 is the total number of days in a year;

60 is the male life expectancy, and 67 is the female life expectancy in South Africa.

Therefore, for a risk assessment calculation, we need the adjusted lifetime chronic daily intake (CDI_adj._), taking in to account the life expectancy for a female and a male South African adult resident. In Equation (4), we calculated the average CDI_adj._, assuming a lifetime daily dose intake.
(4)$${\mathrm{CDI}}_{\mathrm{adj}}=\frac{CDI\;\left(30\;years\;average\;dose\right)}{life\;expectency\;in\;days\;}$$

We assume that the average chronic daily adjusted dose over a lifetime amongst female and male adults will better simplify the risk assessment calculation, as in Equation (4).

#### 2.4.3. Toxicity Assessment and Risk Characterization

Risk characterization is the last step in the risk assessment, which provides information on the hazard status of a contaminant or pollutant [72]. For both carcinogenic and non-carcinogenic effects, the use of a inhalation reference concentration (RfC) assists in determining the health risks that are associated with an exposed population. For carcinogenic pollutants (such as benzene), the use of the slope factor can be used to estimate the relative risk. Furthermore, the use of the inhalation reference concentration was based on toxicological/occupational epidemiology studies, focusing on several health outcomes, such as cellular necrosis. In summary, the inhalation reference concentration (RfC) is an estimated daily human inhalation exposure that is suggested to not cause a health effect in a lifetime [46,47,73].

A lifetime inhalation dose of BTEX was determined, based on the absolute lymphocyte count (ALC) at the adjusted benchmark concentration (BMCL) of 8.2 mg/m^3^. The inhalation benzene lifetime exposure was therefore calculated, using the benchmark dose modeling, and it was found to be 0.03 mg/m^3^. The value of 0.03 mg/m^3^ was therefore described to be the RfC for benzene [72]. The non-carcinogenic effects of the TEX inhalation reference concentration for each pollutant was used to calculate the hazard quotient, as in Table 2 [74,75,76].

Since benzene is the only confirmed human carcinogenic (category A) pollutant amongst the BTEX pollutants, the slope factor was used to calculate the cancer risk. We have adopted the methodology for calculating the cancer risk, using the slope factor from previous similar studies [44,45,46,73,77]. It must be noted that there is no threshold for carcinogenic compounds. Therefore, the use of reference levels is used as a guide, to probably support a decision. In our study, we used both designated cancer severity indicators for one case: 1 × 10^−4^and 1 × 10^−6^ [60,78].

For carcinogenic pollutants, it must be noted that there is no safe threshold; therefore, the risk characterization followed was similar to the method that was described by the USEPA’s Risk Assessment Guidance for Superfund [62]; We thus calculated the risk of cancer by using Equation (5) [73]:CR= CDI_adj_ × SF(5)
where:

SF is the slope factor for carcinogenic pollutant (0.0273);

CR is the carcinogenic risk; and 

CDI_adj._ is the cumulative lifetime adjusted dose (Equation (4)) over an estimated exposure in a lifetime of 60 or 67 years for male and female adult, respectively.

Therefore, a cancer risk >1 × 10^−6^ and 1 × 10^−4^ means that there are carcinogenic effects of concern, while a cancer risk <1 × 10^−6^ and <1 × 10^−4^ means that there is a designated cancer severity indicator.

For non-carcinogenic pollutants, a hazard quotient (HQ) was used to estimate the potential health risk of the dwellers. Where a HQ value is greater than one, it is regarded as a hazardous exposure; a HQ value of less than one means that there is a low probability of developing the associated health effects. In Equation (6), the procedure for calculating HQ is shown.
(6)HQ=CDIadj.((mg/kg)/day)/(RfC(mg/m^3 )×20m3/(70 kg))
where:

HQ is the hazard quotient;

CDI_adj._ is the cumulative intake dose;

R*f*C is the reference;

20 m^3^ is the default value for the average adult daily air volume; and

70 kg is the average body weight for a male adult, while 60 kg will be used for female adult. 

### 2.5. Quality Control

All monitoring instruments were maintained and operated according to the manufacturer’s instructions, and returned to the suppliers for factory calibration at prescribed intervals. Before each test, the gas probes were cleaned by the means of compressed air, to remove any residue from prior tests, which might negatively affect the next results. All monitoring instruments were zero-checked, according to the manufacturer’s instructions, before monitoring/sample collection.

The GC955 instrument was tested in with accordance the EMC directive 89/336/EMC, test specification EN 50081–1:1991 and EN 50082–2: 1994. The monitoring instruments were calibrated before use (calibration was done in the range of 0 to 10 ppb). Quality control checks were conducted during or after the monitoring campaign, and a correction factor of 2 ppb and 4 ppb for benzene and toluene, respectively, were used, to counter systematic under-sampling of the instrument.

Background concentrations were accounted for, as BTEX from outside the testing facility could possibly infiltrate the testing laboratory and contribute to the final concentration readings. The instrument was run for 30 min before the three-hour testing duration, and the background concentrations were calculated, using Equation (7).
(7)$$C_{combustion}=\;C_{activity}-\;C_{without}\;$$
where:

the *C_combustion_* is the final concentration*;*

*C_activity_* is the actual sample collected while the BTEX generating activity was taking place + background concentration;

*C_without_* is the concentration of BTEX obtained in the absence of the activity under investigation.

In experimental studies, the use of the equipment, which are accurately calibrated, is an important quality control feature, and it assists in the reduction of the uncertainty of the dataset. Trial runs before the actual tests might help in the identification of instruments malfunctioning, and detection signal faults.

## 3. Results and Discussion

### 3.1. BTEX Concentration under Laboratory Conditions

The results from the coal combustion brazier under a laboratory-controlled environment are presented herein. In Figure 2, the time aggregates concentration for each BTEX compound is presented as an average concentration for the specified time (15, 45, and 120 min). Using a 3-h average concentration, benzene was the lowest emitted VOC, while ethylbenzene was found to be the most highly emitted pollutant throughout the combustion cycle. From the results, it was shown that the relative concentration of the BTEX species were consistent throughout the entire burn cycle of the three-hour period.

Benzene and ethylbenzene concentration steadily increases, as the combustion process progresses. The minimum concentration, as depicted from Figure 2, is associated with the first 15 min of the combustion. Contrary to benzene and ethylbenzene, the concentrations of toluene and xylene were the highest at 45 min and 120 min, respectively. The observed BTEX profile reported in our study was similar to the one presented in the study by [29]. However, the observed differences may require additional statistical analyses, in order to provide more details on the concentration variation at different time intervals. Unfortunately, the differences in BTEX concentration at different time aggregates were not within the scope of the current project. The implication of this finding indicates for the first time in the South African domestic sector that the determination of domestic coal combustion as might be an important source of BTEX in indoor air spaces.

In Table 3, BTEX near-field room concentrations are presented for replicates of three experiments as averages over a 3-h burn cycle. Benzene concentration ranged from 857–942 µg/m^3^, with a mean of 919 µg/m^3^ over a three-hour burn cycle. The benzene concentration observed in our study varied from those conducted in India, where the concentrations have ranged from 44–167 µg/m^3^ [50]. However, in the latter study, the emissions of benzene were associated with kerosene burning, which is different from our present study. Lower values of indoor benzene concentrations were also reported in several other studies where the concentration ranged from 0.7–7.2 µg/m^3^ [79,80,81]. In the Hong Kong Special Administrative Region of China, similar low benzene indoor levels were reported, which were mainly associated with vehicular emissions at 0.5–4.4 µg/m^3^ [30,82]. However, studies conducted in petrol refineries reported that concentrations for benzene varied between 12–17,000 µg/m^3^, with the highest exposure concentrations being mainly from refinery workers working in indoor environments [83,84,85].

Toluene, ethylbenzene, and xylenes (TEX) results are comparable with several studies conducted elsewhere; however, most of these studies were conducted in occupational settings [34,59,70]. The ethylbenzene concentration measured in our study was higher than the concentration reported elsewhere [34]. In the study by [34], the focus was on an occupational setting, which is suggested to be highly contaminated, relative to the residential environment. In this light, it can be seen that the exposure in a residential environment might be higher than in occupational settings, especially where coal burning is used as a primary energy source. In summary, our toluene, ethyl benzene, and xylenes were not the highest concentrations reported in the field. Despite the TEX results being found to be lower compared to the highest reported concentrations in other studies, it is suggested that they may present several health effects, even at lower concentrations [86,87,88].

In Table 4, we investigated a percentage contribution of individual BTEX compounds. From the total BTEX indoor air concentration, benzene was found to have contributed less at 6%, while ethyl benzene was the highest, at 29%. Fairly comparable percentage contributions between P-xylene and O-xylene were observed at 26 and 25, respectively. However, despite benzene being the least quantified VOC, it is worrying, given its hazardous nature to human health. Toluene was found to be the lowest contributed VOC amongst the TEX, at 14%.

### 3.2. Potential Health Risk Analysis of BTEX

Results presented in Table 5 and Table 6 depict the carcinogenic and non-carcinogenic risks of BTEX exposure from domestic coal burning for adult females and males, respectively. The determination of risks associated with BTEX were achieved when using the cancer risk for the carcinogenic compound (benzene), while the non-carcinogenic effects of TEX were determined by calculating the hazard quotient, as shown in Equations (5) and (6), respectively. The cancer risk for adult females and males were determined to be 1.2 × 10^−4^ and 1.1 × 10^−4^, respectively. The cancer risk for females was found to be higher than that of males. This finding suggests that women will be more vulnerable than men, even though the exposure concentration is the same. As shown in Table 5, the cancer risk for women suggests that 120 people will be at risk of cancer per million people in the exposed population. Furthermore, in Table 6, the results show that 110 men per million people exposed will be at risk of carcinogenic health effects. In both exposure scenarios (male and female), the cancer risk was found to be higher than the acceptable risk levels of 1 × 10^−6^ and 1 × 10^−4^.

The cancer risks for adult females and males were determined with reference to the female/male body weights (default value from USEPA, 2010) and life expectancies [65], as in Table 1. The cancer risks in adult females and males were found to be 120- and 110-fold higher than the designated cancer severity indicator of 1E^-6^, respectively. These findings confirm those reported by [89,90], where 17% of premature lung cancer deaths in adults were found to be attributable to exposure to carcinogens from household air pollution caused by cooking with kerosene or solid fuels, such as wood, charcoal, or coal and the risk for women was higher, due to their role in food preparation.

For non-carcinogenic health effects, a hazard quotient was used to determine the risk. A hazard quotient of greater than 1 was used as a reference value; whereby, a value greater than one indicated a higher probability of contracting a related health effect. For both adult males and females, the hazard quotient was found to be below 1 for the TEX. Toluene indicated the lowest hazard quotient, whilst ethylbenzene was potentially found to have the highest hazard quotient, at a value of 0.05. The results presented in our study indicate that there is a lower probability of non-carcinogenic health effects as a result of exposure to domestic coal combustion technology, as described in this study. 

Despite the non-carcinogenic effects rating a hazard quotient of less than one, this might change significantly, especially in households where coal burning devices are used indoors for longer durations. This includes overstretched winter periods, and prolonged exposure durations, based on activity. For instance, in some households, especially during winter, this type of technology can be used to warm for the entire day time (06:00 to 18:00), and for some part of the night period (18:00 to 21:00). This implies that exposure to TEX from this combustion activity may significantly vary from one household to the other, depending on the case scenarios used.

## 4. Study Limitations

In the absence of field exposure data, the results presented herein had several limitations.

The individual information used for risk assessment are average person default values. As it is known that there are no average people in the world, this might significantly affect the accuracy of risk determination. Individuals vary based on the biological make-up, which might affect parameters such as the breathing rate, and moreover, the exposure scenario. Furthermore, we used average values to overlook the issue of individual susceptibility, which might affect the risk score. In addition, we used a room concentration to assess the risk, assuming that a three-hour exposure at near-field breathing zone is the average exposure duration. The influence of pollutants leakage was not addressed, where there might be loss due to leakages. The BTEX concentration reported in this study was obtained from a laboratory environment, which might vary from field concentrations.

Despite these limitations, this study has shown that exposure to domestic coal combustion pollutants may lead to the risk of carcinogenic effect, while non-carcinogenic effects were found to be unlikely. However, it must be noted that the results presented herein were based on a laboratory experimental study, where several variables that might influence stove to fuel performance were controlled. Such performance determinants includes the stove operational method, the fire ignition method, the coal particle size, the moisture content, and the coal grade [13,53,91,92].

## 5. Conclusion

The study attempted to quantify the BTEX concentration from domestic coal combustion in a brazier, simulating its use in South African informal settlements. Based on the results presented in this study, it can be concluded that domestic coal burning might be a significant source of BTEX in indoor spaces. The results showed a constant concentration of BTEX throughout the combustion cycle of 3 h.

The study further attempted to utilize a breathing zone near-field BTEX concentration, as averaged over a 3-h burning cycle in adult females and males, to estimate the carcinogenic and non-carcinogenic health effects, simulating practices in informal settlements. The cancer risks were found to be 110- to 120-fold higher than the designated cancer severity indicator of 1 × 10^−6^.

The health risk assessment of TEX, through calculating the hazard quotient, was below the reference value of 1; indicating a potentially low exposure to these pollutants, and possibly a reduced risk of the associated health effects. The lessons drawn from this experimental laboratory study indicate the need for further studies in this field to have an improved understanding of exposure scenarios, for informed risk characterization from this source. This study presented the first risk assessment arising from domestic coal burning activities in a laboratory environment, while mimicking field practices that are relevant to the South African situation.

Notably, risk assessment is a comprehensive and iterative process for assessing the relative risk for several exposure scenarios. It must be understood that the risk assessment has several uncertainties, the accuracy of the results depends on the correct risk identification and use of accurate exposure information. Despite all uncertainties, in our studies, we attempted to ensure that the exposure scenarios were accurately defined, which might be used in the future for future studies.

Furthermore, this study has proven that the use of a high ventilated stove and the top-lit updraft might not have a significant effect on the reduction of BTEX, relative to what the technology is reportedly capable of (i.e., the reduction of particulate matter by 80%). However, this study was not intended to carry out a comparative assessment on emissions reduction by using different technologies (stove ventilation and ignition method); such a comparison might be useful in future projects/studies.

In summary, the use of a high ventilated stove and the TLUD ignition method may not be a useful household indoor air pollution intervention, due to the inherent carcinogenic risk. Therefore, other clean energy alternatives may be exploited and be introduced in these settlements, in order to improve indoor air quality.

## Figures and Tables

**Figure 1 ijerph-16-00095-f001:**
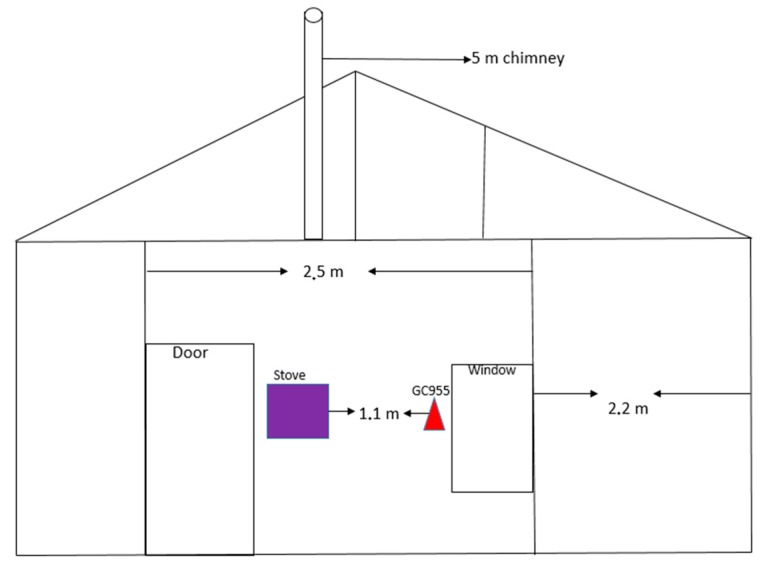
Schematic diagram of a traditional corrugated iron house in a typical South African informal settlement, stove, and GC955 sampling inlet. (Not drawn to scale).

**Figure 2 ijerph-16-00095-f002:**
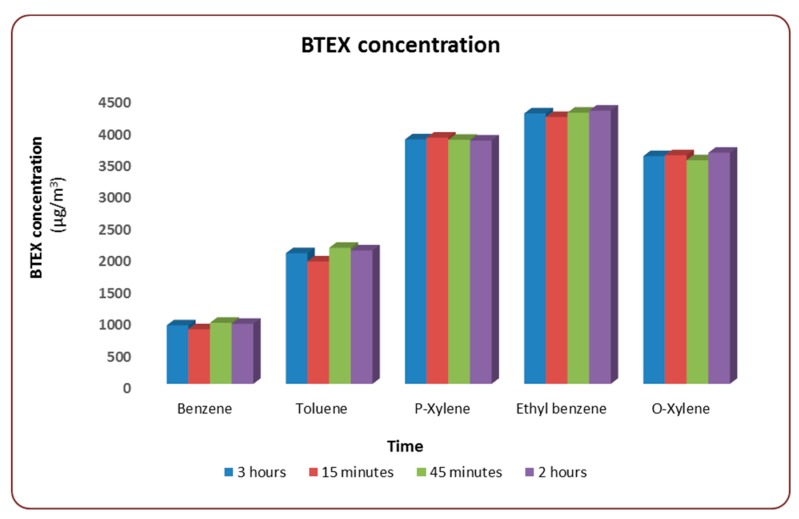
Time series of benzene, toluene, ethylbenzene, and xylenes (BTEX) concentration for a 3-h combustion cycle.

**Table 1 ijerph-16-00095-t001:** Summary of the exposure scenario factors and values used in this study.

Parameter	Description	Value	Unit
C	Room concentration	-	mg/m^3^
IR	Inhalation rate	20	m^3^/day
BW	Body weight	70 males/ 60 kg females	kg
ED	Exposure days	92 (3 h per day)	Days/year
YE	Years of exposure	30 (Residential)	Years
AT	Years in lifetime	60 male/67 female	Years

The default inhalation rate, body weight, and residential exposure from U.S. Environmental Protection Agency (USEPA) [61], while the male and female years in life were adopted from [64].

**Table 2 ijerph-16-00095-t002:** Benzene slope factor, and toluene, ethylbenzene, and xylenes (TEX) inhalation reference values.

Chemical	Inhalation Reference Concentration (R*f*C)	Inhalation Slope Factor (SF)
	(mg/m^3^)	(mg/kg/day)^−1^
Benzene	0.03	0.0273
Toluene	5	N/A
Ethylbenzene	1	N/A
O-xylene	0.1	N/A
P-xylene	0.1	N/A

**Table 3 ijerph-16-00095-t003:** Time-weighted average BTEX room concentrations.

Duration	Benzene	Toluene	P-Xylene	Ethylbenzene	O-Xylene
	(µg/m^3^)	(µg/m^3^)	(µg/m^3^)	(µg/m^3^)	(µg/m^3^)
	*n* = 3	*n* = 3	*n* = 3	*n* = 3	*n* = 3
15 min	857 ± 32.40	1922 ± 127.5	3864 ± 48.33	4189 ± 87.11	3589 ± 48.74
45 min	958 ± 5.73	2137 ± 27.04	3831 ± 15.12	4257 ± 31.26	3510 ± 13.66
2 h	942 ± 13.36	2095 ± 36.59	3819 ± 9.60	4288 ± 91.51	3628 ± 9.42
3 h Average concentrations	919 ± 44	2051 ± 93	3838 ± 19.04	4245 ± 41.13	3576 ± 49

**Table 4 ijerph-16-00095-t004:** Percentage contribution of each BTEX pollutant, averaged over a 3-h burn cycle ignited in a high ventilated stove (HIGH) and ignited using the top-lit updraft method (TLUD).

Pollutant	Ignition	Concentration	Contribution
	Stove Ventilation	(µg/m^3^) *n* = 3	%
Benzene	TLUD	919 ± 44	6
	HIGH		
Toluene	TLUD	2051 ± 93	14
	HIGH		
P-Xylene	TLUD	3838 ± 19.04	26
	HIGH		
Ethyl benzene	TLUD	4245 ± 41.13	29
	HIGH		
O-Xylene	TLUD	3576 ± 49	25

**Table 5 ijerph-16-00095-t005:** Carcinogenic and non-carcinogenic risks for adult females.

Pollutant	Average concentration	CDI_year_	CDI_30 year_	CDI_adj._	CR	HQ	CR/10^6^	CR/10^4^
	µg/m^3^	mg/kg/day	mg/kg/day	mg/kg/day				
Benzene	919	0.0097	1.06 × 10^2^	4.32 × 10^−3^	1.2 × 10^−4^	N/A	120	1
Toluene	2051	0.0215	2.36 × 10^2^	9.64 × 10^−3^	N/A	0.001	N/A	N/A
P-Xylene	3838	0.0403	4.41 × 10^2^	1.73 × 10^−2^	N/A	0.050	N/A	N/A
Ethylbenzene	4245	0.0446	4.88 × 10^2^	2.00 × 10^−2^	N/A	0.006	N/A	N/A
O-Xylene	3576	0.0376	4.11 × 10^2^	1.68 × 10^−2^	N/A	0.049	N/A	N/A

CDI_year_: the estimated daily intake corresponding to an annual dose; CDI_30 year_: cumulative average 30-year dose; CDI_adj._ is the cumulative intake dose; CR: carcinogenic risk; HQ: hazard quotient.

**Table 6 ijerph-16-00095-t006:** Carcinogenic and non-carcinogenic risks for adult males.

Pollutant	Average Concentration	CDI_year_	CDI_30 year_	CDI_adj._	CR	HQ	CR/1E^6^	CR/1E^4^
	µg/m^3^	mg/kg/day	mg/kg/day	mg/kg/day				
Benzene	919	0.0083	9.06 × 10	3.70 × 10^−3^	1.1E × 10^−4^	N/A	110	1
Toluene	2051	0.0185	2.02 × 10^2^	8.27 × 10^−3^	N/A	<0.001	N/A	N/A
P-Xylene	3838	0.0345	3.78 × 10^2^	1.55 × 10^−2^	N/A	0.045	N/A	N/A
Ethyl benzene	4245	0.0382	4.18 × 10^2^	1.71 × 10^−2^	N/A	0.005	N/A	N/A
O-Xylene	3576	0.0322	3.52 × 10^2^	1.44 × 10^−2^	N/A	0.042	N/A	N/A

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
