# Peer review of "Risk Assessment of Benzene, Toluene, Ethyl Benzene, and Xylene Concentrations from the Combustion of Coal in a Controlled Laboratory Environment"

_ijerph, 2018, doi:10.3390/ijerph16010095_

Reviewer 1 Report

I found this paper had some useful results and I didn’t see any major errors. However, I think further editorial review is necessary as I found typos, garbled sentences and then I gave up halfway through the manuscript. My biggest concern relates to the stove that was used. It’s not clear why this model was chosen or if it is representative of what is used in common practice. So, it’s not clear how relevant the results are. This paper would have been much more useful if it included a comparison, using the same measurements and risk assessment methods, for stove(s) in current use. I think this paper could be published after correcting the minor errors and a comparison is made to stoves currently used. 

Minor comments

In the caption to Figure 1, GC995 is not identified. Don’t make the reader have to go find it later in the text.

l. 145 – operates

l. 174 – Because there is an international audience, it might be helpful to state that value as a reserve currency, e.g., dollars, pounds, or euros.

l. 174 – resides. Sentence is garbled and needs to be rewritten.

l. 188 – population index needs explication or replacement.

l. 321 – relative concentrations of BTEX species might be better

Figure 2 – emissions concentrations is a confusing term. Drop emissions as it’s clear from the text that you’re talking about concentrations during the combustion cycle.

l. 334-345 – Results of comparisons are presented better in a table showing concentration ranges, fuel type and setting.

l. 347 - these

l. 346-350 – it’s impossible to follow what concentrations were in (80) and why that’s relevant.

l. 349-350 – sentence needs expansion and explanation.

Author Response

I would like to thank the reviewer of International Journal of Environmental and Public Health – for the detailed and insightful reading of my article document, and the valuable suggestions for its improvement. 

Reviewer 2 Report

Daniel Masekameni et al assessed the health risk of BTEX concentration from combustion of coal in the controlled laboratory condition in South Africa. The manuscript is suggested to be accepted after major revision. 

Major questions:

1. As mentioned by the authors in Introduction, BTEX risk studies have been studied in countries such as USA, India and China. The obvious question is novelty of the study. Since the health risk should be evaluated by the concentration of BTEX, the risk should be the same no matter it is South Africa or not. Especially, it was studied in the laboratory conditions, the results should be the same. The authors need to further explain the necessity of this study.

2. The author did not compare the results of this study from the that in USA, India and China.

Minor questions:

1.       Typo: P1, line 27 and 29; TEX should be BTEX

2.       Page 11, line 428  TEX should be BTEX

3.       Page 10, line 363 TEX should be BTEX

4.       Page 1, line 26: the authors should rearrange the sentence here because it seemed that the cancer risk for adult males are 1.1-1.2; the cancer risk for adult females are 110-120.

Author Response

I would like to thank the reviewer of International Journal of Environmental and Public Health – for the detailed and insightful reading of my article document, and the valuable suggestions for its improvement. The comments as presented by the reviewer mainly highlight corrections to be applied to the document regarding use of acronyms, addition of units, correcting missing text and justification of arguments, as a whole.

Round  2

Reviewer 1 Report

I have only minor comments for the authors. Once these are taken care of, the paper can be published.

Figure 2 – BTEX emissions concentrations.

the selection of high ventilated stoves was based on local studies which proven that the use of high ventilated stove lit with TLUD leads to the reduction of emissions.

I suspect the reviewer missed this.

No, the reviewer didn’t miss this. The questions was reduction of emissions – compared to what? Many readers are not familiar with the stoves in use in South Africa and so it would want be useful to at least put in a reference to another paper that addresses BTEX exposure from these other stoves. The conclusion of this paper is important and needs further support if it is going to be used to address indoor air quality in South Africa.

Otherwise the response to this work very well might be –“so what?”

Author Response

My sincere appreciation to the reviewer. Significant contribution has been made to this article and in that I will like to send my appreciation to the reviewer. I agree with all comments/ suggestions and I have tabulated my response as attached.

Reviewer 2 Report

Concerns have been addressed.

Author Response

My sincere appreciation to the reviewer. Significant contribution has been made to this article and in that I will like to send my appreciation to the reviewer.